# Rapeseed Meal and Its Application in Pig Diet: A Review

Hao Cheng [1], Xiang Liu [1], Qingrui Xiao [1], Fan Zhang [1], Nian Liu [1], Lizi Tang [1], Jing Wang [1], Xiaokang Ma [1,*], Bie Tan [1], Jiashun Chen [1] and Xianren Jiang [2]

1   College of Animal Science and Technology, Hunan Agricultural University, Changsha 410128, China; chenghao19970316@163.com (H.C.); liuxiang1138120110@163.com (X.L.); 15773772597@163.com (Q.X.); fzhang4056@163.com (F.Z.); ln18874045069@163.com (N.L.); t1523102483@163.com (L.T.); jingwang023@hunau.edu.cn (J.W.); bietan@hunau.edu.cn (B.T.); llh0920@163.com (J.C.)
2   Key Laboratory of Feed Biotechnology of Ministry of Agriculture and Rural Affairs, Institute of Feed Research, Chinese Academy of Agricultural Sciences, Beijing 100081, China; jiangxianren@caas.cn
*   Correspondence: maxiaokang@hunau.edu.cn

**Abstract:** Rapeseed is the second largest plant protein resource in the world with an ideal profile of essential amino acids. Rapeseed meal (RSM) is one of the by-products of rapeseed oil extraction. Due to the anti-nutritional components (glucosinolates and fiber) and poor palatability, RSM is limited in livestock diets. Recently, how to decrease the anti-nutritional factors and improve the nutritional value of RSM has become a hot topic. Therefore, the major components of RSM have been reviewed with emphasis on the methods to improve the nutritional value of RSM as well as the application of RSM in pig diets.

**Keywords:** rapeseed meal; pig; plant protein; livestock

## 1. Introduction

The imbalance between supply and demand of protein feed resources in developing countries has been increasing acutely. The feed production industry is facing a demand challenge. Therefore, the effective utilization of feed resources has attracted more attention. The United States Department of Agriculture states that rapeseed, one of the excellent plant protein resources, is the second-largest protein resource in the world with a global output of 73.09 million tons in 2017, which is only less than soybean [1,2]. Data from the National Bureau of Statistics shows that the yield of rapeseed in China has reached 14.05 million tons in 2020, which has become one of the major protein crops (Figure 1). The rapeseed oil is mainly utilized for human diet and the rapeseed meal (RSM) is a co-product which is commonly used as a protein source in animal diets after the oil has been extracted [3].

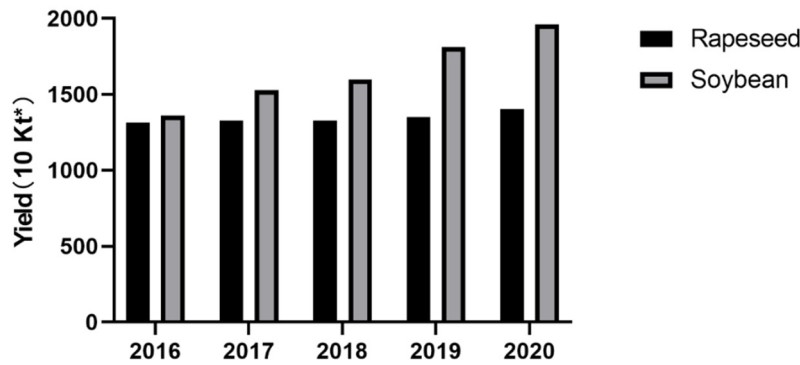

**Figure 1.** Soybean and rapeseed yield from 2016–2020 in China; Kt* = kiloton.

RSM is abundant in crude protein and has well-balanced essential amino acids, which is very close to the published values of soybean protein from the United Nations Food and Agriculture Organization [4]. However, the application of RSM in nutrition is limited because of the existence of anti-nutritional factors and toxic substances.

In recent years, the use of RSM in feeds for swine has increased with the development of low glucosinolate rapeseed by Canadian and European plant breeders [5]. Meanwhile, many technologies have been developed to improve the nutrient digestibility of RSM in growing pigs efficiently, including fermentation and hull-removing [6,7]. However, there are few systematic reviews of the application of RSM in pig diets. To maximize the benefits of RSM, it is vital to understand its nutritional characteristics and develop technologies to enhance its utilization cost-effectively [8]. Therefore, this review summarized the limitation of the utilization of RSM in animal diets from the nutritional value standpoint, and some improvement methods to enhance the quality of RSM.

## 2. Composition and Characteristics of Rapeseed Meal

### 2.1. Crude Protein and Amino Acids

As a raw material of protein feed, the content of crude protein (CP) in RSM is usually between 35% and 40% (Table 1). The protein content of RSM is affected by rapeseed type, growing environment, fiber content and so on. It has been found that the RSM produced from yellow-seeded Brassica juncea contained, on a dry matter basis, higher crude protein than that produced from traditional black-seeded Brassica napus canola and yellow-seeded Brassica juncea [9]. Earlier studies have demonstrated a sharp negative relationship between protein and dietary fiber content in meals [10] and crude protein content of RSM is significantly increased after the shell was removed [11].

**Table 1.** Composition of crude protein and essential amino acids of rapeseed meal and soybean meal (as fed basis).

|  | Rapeseed Meal | Double-Low Rapeseed Meal | Soybean Meal |
|---|---|---|---|
| Crude protein (%) | 37.6~38.2 | 39.4~43.6 | 43.8~49.9 |
| Arginine (g/kg) | 20.6~22.1 | 20.8~24.1 | 34.9~37.8 |
| Histidine (g/kg) | 10.0~10.1 | 10.4~12.0 | 12.1~13.2 |
| Isoleucine (g/kg) | 14.6~15.3 | 13.8~15.6 | 21.5~27.8 |
| Leucine (g/kg) | 26.6~27.0 | 25.4~27.6 | 36.6~39.2 |
| Lysine (g/kg) | 17.2~19.5 | 19.4~24.1 | 29.9~32.2 |
| Methionine (g/kg) | 7.4~7.6 | 7.6~9.7 | 6.0~6.9 |
| Phenylalanine (g/kg) | 15.1~15.3 | 12.2~13.6 | 23.5~30.0 |
| Threonine (g/kg) | 17.5~17.6 | 17.6~19.1 | 18.9~20.3 |
| Tryptophan (g/kg) | 5.0~5.1 | 4.6~5.4 | 6.6~7.5 |
| Valine (g/kg) | 18.6~19.7 | 21.5~23.8 | 22.4~26.7 |
|  | Mosenthin et al. [12] | Li et al. [13] | Banaszkiexicz [14] |

The protein from RSM is less digestible than that from soybean meal (SBM), while the amino acids balance of RSM is similar to that of SBM [15]. Compared with SBM, the concentration of lysine in RSM is lower, while the concentration of sulfur-containing amino acid in RSM is higher [16]. Therefore, it is better to mix RSM with SBM with a reasonable ratio when used for livestock diet due to the complementarity of amino acids content. Salazar-Villanea et al. found that toasting time has no influence on the CP content, but decreased the content of lysine, alanine, and glutamic acid and there was a significant correlation between the rate of protein hydrolysis and lysine [17]. It is worth of noting that heat treatment and other steps during processing may change the physical and chemical structure of protein while having no effects on the nitrogen content. Therefore, the content of available protein and amino acids in RSM can't be determined by the crude protein content while the content of lysine might be a better parameter to indicate the protein quality in RSM.

### 2.2. Crude Fat

RSM is the residue of rapeseed oil extracted by different processing techniques. The concentration of crude fat varied in dry matter of RSM from 1.42% to 14.55% with a mean of 7.78% [18]. The type of rapeseed, impurity content, processing technology, planting conditions in different producing areas, such as climate, and other factors can all affect the crude fat content in RSM [19]. Theodoridou and Yu found that the concentration of crude fat was higher from the brown-seeded Brassica napus canola press cake than from the yellow Brassica juncea and the browm-seeded Brassica napus canola meal [20]. During the drying, softening, rolling, steaming, and reserving processes, the value of the crude fat content of RSM as fed basis changes with the change of moisture content. Bojanowska reported that in comparison with fresh samples, significant differences in the crude fatcontent were observed in RSM after storage with increased saturated fatty acids and decreased unsaturated fatty acids [21]. However, as the utilization of RSM in swine diets is limited, relatively few studies are available on the crude fat content of RSM, and further research is needed.

### 2.3. Carbohydrate

The carbohydrates account for about one-third of the RSM and are mainly composed of monosaccharide, sucrose, oligosaccharide, starch, and NSP (non-starch polysaccharide). The content and structure of NSP in RSM vary with the type of rapeseed. Slominski and Campbell reported that the NSP content of Brassica campestris meal is between 16–22% while Pustjens et al. found the NSP content of Brassica napus meal is 24% [22,23]. In rapeseed, low-molecular-weight carbohydrates are mainly found in cotyledons while high-molecular-weight carbohydrates are mainly found in hulls [24]. Hulls contained 2.9% of the oil, 11.2% of protein, 73% of NDF (neutral detergent fiber), 80% of ADF (acidic detergent fiber) and 95% of ADL (lignin) and 6.0% of the glucosinolates of the whole rapeseed and RSM contained 48.3% protein (dry basis), 10.8% of NDF, 6.6% of ADF, and 0.5% of ADL after completely dehulled [25] The RSM has a limited utilization in swine diets due to the highly indigestible carbohydrates. Some studies have shown that rapeseed types and processing technologies affect the content of crude fiber in RSM, such as the dehulled RSM containing less crude fiber [19,25]. Along with the rapid development of alternative protein resources, it is necessary to develop some methods to improve the nutritional value of RSM.

### 2.4. Minerals and Vitamins

RSM is a relatively rich source of minerals including calcium, phosphorus, potassium, iron, zinc, and selenium which are important minerals for pigs [26]. Compared with SBM, RSM contains higher calcium and phosphorus contents [27]. In terms of vitamin composition, RSM contains a great amount of vitamin B such as biotin, folic acid, niacin, riboflavin and thiamin and vitamin E [28]. Moreover, RSM is rich in phenolic compounds and other bioactive compounds such as tocopherols and choline [29]. These studies above have shown that RSM has the potential to be a good protein feed resource.

### 2.5. Anti-Nutritional Factors

The nutritional value of RSM is close to that of SBM, but RSM is limited in practical appliances because it contains harmful substances and anti-nutritional factors, such as glucosinolates, phytic acid and fiber.

#### 2.5.1. Glucosinolates

Glucosinolates are a large group of sulphur-containing secondary plant metabo-lites which are widely found in cruciferous plants and vegetables. The major forms of glucosinalates in RSM are gluconapin (3-butenyl), glucobrassicanapin (4-pentenyl), progoitrin (2-hydroxy-3-butenyl), gluconapoleiferin (2-hydroxy-4-pentenyl), and glucobras-sicin (3-indolymethyl) [30–32]. Generally, the natural and non-hydrolyzed forms of glu-cosinolates are not harmful to animals or humans. But, glucosinolates will produce glucosi-

nolate derivatives including nitriles, thiocyanates, isothiocyanates and 5-vinyloxazolidine-2-thione after being hydrolyzed which have been reported to interfere with iodine uptake by thyroid gland and induce goitrogenic effects in humans [33]. Moreover, gluconapin will affect the palatability of RSM due to its' pungent taste and the degradation product of progoitrin can be harmful to the health and function of thyroid [34,35]. Glucosinolates are the major anti-nutritional factor in RSM that reduces animal performance by affecting palatability of RSM and feed intake, impairing thyroid, liver, and kidney function and reducing feed utilization efficiency [36,37]. Velayudhan et al. demonstrated that replacing 33–66% of SBM with expeller extracted RSM (containing 1.90–2.78 μmol/g glucosinolates) in growing pig diets increased thyroid weight, changes in thyroid hormones, and linearly decreased the average daily feed intake [38]. Seneviratne et al. found that the tolerant level of glucosinolates in grower-finisher pigs was 1.70–3.40 μmol/g [39]. Whereas, Landero et al. reported that body weight gain, feed intake, and feed efficiency were reduced in weaned pigs even at the lowest dietary inclusion (6%) of juncea canola meal containing 0.65 μmol/g glucosinolate in wheat-based diets, which indicates a high sensitivity of young pigs to glucosinolates [33]. Many studies have shown that as the early life production stressor, weanling not only changes structural and functional intestine, but also contributes to an intestinal inflammatory status which impairs gut barrier function [40–42]. We speculate that the fact young pigs are more sensitive to glucosinolates than older pigs may due to the immature gastrointestinal tract function [43].

### 2.5.2. Phytic Acid

Phytic acid (myoinositol hexaphosphoric acid) is the stored form of phosphorus in grains, legumes, nuts, and seeds [44]. Phytic acid is considered an anti-nutritional factor because it forms insoluble complexes by combining with proteins and several minerals (Zn, Ca, and Fe). This interaction may effect changes in protein structure and protein solubility which render them unavailable for intestinal absorption in humans and animals [45].

### 2.5.3. Tannin

Food tannins are polyphenolic compounds with molecular weights of 500–3000 Daltons, which can be divided into condensed or hydrolysable tannins and most of the tannins in rapeseed are condensed tannins [46]. Tannins can reduce the bioavailability of the nutrients due to the ability to form indigestible and bitter-tasting complexes with proteins [47].

## 3. Improvements to the Nutritive Values of Rapeseed Meal

In order to improve the overall nutritive value of RSM, enhance the utilization rate of RSM in pig diets and reduce the waste of protein resources, it is necessary to detoxify RSM and improve the processing techniques. At present, the nutritional value of RSM can be enhanced by optimizing processing conditions, reducing fiber content, and plant breeding strategy.

### 3.1. Optimize Processing Conditions

The processing technology of RSM has a great influence on the quality of RSM. Woyengo et al. reported that the expeller-extracted canola meal had higher digestible amino acid, digestible energy, and metabolic energy contents than the solvent-extracted canola meal [48]. Moreover, heat treatment in the processing also affects the nutritional value of RSM. For cold-press canola cake, application of heat to the barrel of the press during oil extraction increased the apparent ileal digestibility and apparent total tract digestibility of energy of cold-pressed canola cake [49]. While for hot-press, medium seed conditioning temperature resulted in the highest ileal digestible energy and apparent metabolic energy compared with low and high temperature [50]. It can be seen from the above that proper temperature adjustment during the pressing process can improve the nutritional value of RSM. The optimization of the process requires the joint efforts of different feed and oil

enterprises to detect the nutritional value of oil and RSM, and then find the best control point of processing technology.

*3.2. Reduce Fiber Content*

In recent decades, various approaches including breeding for low-fiber and dehulling of seed have been undertaken to reduce the fiber content and improve the nutrient value of RSM [10,51].

Selective breading of rapeseed is not only aimed at reducing glucosinolate content but also decreasing fiber content [51]. Earlier research has demonstrated that the yellow-seed canola contains more protein and sucrose, less fiber and similar amounts of oligosaccharides and minerals compared with brown-seeded [52]. Slominski et al. also demonstrated that RSM derived from yellow-seeded Brassica napus canola contained more protein (49.8 vs. 43.8% DM), more sucrose (10.2 vs. 8.8% DM), and less total dietary fiber (24.1 vs. 30.1% DM) in comparison with conventional black-seeded counterpart [10].

Removing the hull of rapeseed before oil extraction is a good method to reduce fiber content. Kracht et al. studied the effects of removing the hull of rapeseed on the composition of RSM and rapeseed cake before the rapeseed is pressed and found that removing the hull of rapeseed significantly decreased the crude fiber content in RSM and rapeseed cake by approximately 40%, the neutral detergent fiber content by 28% and 39%, and the acid detergent fiber content by 35% and 39% in RSM and rapeseed cake, respectively [53]. Additionally, the hull of RSM could be removed by sieving particle size or by air classification according to density after production. It has been observed that the acid detergent fiber and the neutral detergent fiber were reduced by 31.9% and 29.5% in the light-particle fraction and were enriched by 16.5% and 9.0% in the heavy-particle fraction compared with parent canola meal [54]. Meanwhile, the air classification of canola meal increased its energy and acid amino digestibility in the light-particle fraction because of the reduced dietary fiber content. These results are in agreement with those reported by Hansen et al., who reported that sieving and air classification could remove the fiber from RSM and the reduced fiber content and increased crude protein content resulted in a higher digestibility of crude protein and amino acids [55].

Taken together, breeding for low-fiber rapeseed and removing the hull of rapeseed could reduce fiber content and enhance nutritive value, which have been revealed as effective methods to improve the nutrient value of RSM.

*3.3. Biological Method to Improve Utilization of Rapeseed Meals*

3.3.1. Enzymic Method

A relatively abundant literature has described the application of dietary enzymes (phytase, protease and carbohydrases) on RSM for the purpose of facilitating phosphorus, protein, and energy utilization for animal feed (Table 2).

Generally, phytases are known to improve phosphate and mineral uptake in animals which can't metabolize phytate [68]. Maison et al. reported that the addition of microbial phytase in growing pig diet improved digestibility of dietary phosphorus in RSM [58]. Potocka et al. found similar results, indicating that addition of phytase additives in growing and finishing pig diets could improve phosphorus and calcium digestibility [57]. Moreover, Rodrigues et al. reported that compared with a simple extraction process under similar conditions, pretreatment of RSM with phytase combined with alkaline extraction could enhance protein extraction yield [69]. It can be seen from the above that in addition of phytase can improve the utilization of RSM. However, the effect of adding phytase on growing pig diets is also different due to the difference of the source, dose and method of phytase application.

The use of non-starch polysaccharide enzymes, including xylanase, glucanase, cellulase, have been extensively studied in pig diet. It has been found that cellulase and alkaline feeding considerably changed the microbiome, as well as improved the overall degradation of RSM [70]. Li et al. also reported that multi-enzyme (cellulase, xylanase, glucanase,

and protease) supplementation increased the crude protein digestion and all amino acids absorption and enhanced fiber degradation in double-low rapeseed co-products fed to pigs [60]. Similarly, Long and Venema also found that cellulose and pectinases could change microbial community composition, increase the abundance of microbial fiber-degrading enzymes and pathways, and increase acetic acid, propionic acid, butyric acid, and SCFA production [62]. In addition, non-starch polysaccharide enzymes could also reduce the viscosity of chyme, degrade plant cell walls and reduce the proliferation of pathogenic microorganisms in the intestine [71].

**Table 2.** Biological method application of rapeseed meal on pig.

| Method | Treatment | Animal | Main Results | Reference |
|---|---|---|---|---|
| Enzymic method | Aspergillus ficuum phytase | Weanling pig | Increase digestibilities of P and Ca and improve bone structure | Zhang et al. [56] |
| | Quantum Blue phytase | Growing and finishing pig | Improve P and Ca digestibility, and reduce P excretion | Małgorzata et al. [57] Maison et al. [58] |
| | Carbohydrases and phytase | Sow | Reduce the body weight loss and improve P digestibility post-farrowing | Velayudhan et al. [59] |
| | Cellulose, xylanase, glucanase and protease | Growing and finishing pig | Increased the standardized ileal digestibility of crude protein and all amino acids and enhanced fiber degradation | Li et al. [60] Torres et al. [61] |
| | Cellulose and pectinase | Growing pig | Change microbial community and increase the abundance of microbial fibre-degrading enzymes and pathways | Long and Venema [62] |
| | Cellulase, pectinase, amylase, protease and phytase | Gestating and lactating sow | Improve the standard ileal digestibility of amino acid | Velayudhan et al. [63] |
| Microbiological fermentation | *Bacillus subtilis* and *Lactobacillus fermentum* | Weaned piglet | Reduce the incidence of diarrhea and improve the gut microbiota | Czech et al. [64] |
| | *Aspergillus niger 41258* | Growing barrow | Increase P digestibility and digestible amino acid content and decrease P excretion | Shi et al. [65] |
| | *Lactobacillus* | Pregnant sow | Improve the structure and mechanical properties of compact bone in offspring | Tomaszewska et al. [66] |
| | *Lactobacillus*, cellulose and pectinase | Growing pig | Increase reducing sugars and reduce the glucosinolate, total short-chain fatty acid and acetic acid content | Zhu et al. [67] |

### 3.3.2. Microbiological Fermentation

Microbial fermentation RSM refers to the use of complex microbial community and complex microbial enzyme system to ferment and decompose the toxic components and enzymatic hydrolysis products of RSM to achieve the purpose of detoxification. Microbial fermentation has an excellent detoxification effect on glucosinolates in RSM, which is the main anti-nutritional factor in RSM (Table 3). Zhang et al. found that under the fermentation of *Lactobacillus delbrueckii* and *Bacillus subtilis*, the content of glucosinolates in RSM decreased from 64.56 μmol/g to 3.47 μmol/g, and the degradation rate was high as 94.62% [72]. It has been reported that the soluble protein content, lactic acid content and total amino acid content in RSM increased significantly, whereas the glucosinolate content and neutral detergent fiber content decreased significantly after fermentation of *Bacillus licheniformis*, *Yeast* and *Lactobacillus* [73]. The lactic acid could contribute to the acidic taste and denature protein to reduce chewiness. Vig and Walia employed solid state fermentation of RSM and found that the contents of glucosinolates, thiooxazolidones, phytic acid and crude fibre declined by 43.1%, 34%, 42.4% and 25.5%, respectively, following inoculation with *Rhizopus oligosporus* [74].

Taken together, these studies have shown that microbial fermentation not only decreases the content of anti-nutritional material (glucosinolate, phytic acid, and crude fiber), but improves the palatability and the nutrient digestibility of RSM. Moreover, detoxi-

fied RSM by microbial fermentation has the advantages of obvious detoxification effect, mild condition, and the low expenture of process and add the basis for the increase in attention [79].

**Table 3.** Effects of microbiological fermentation on glucosinolates in rapeseed meal.

| Source | Glucosinolates Content (μmol/g) | Treatment | Degradation Ratio | Reference |
|--------|--------------------------------|-----------|-------------------|-----------|
| Canola meal | 9.31 | *Aspergillus sojae* and *Aspergillus icuum* | 30% | Olukomaiya et al. [75] |
| Rapeseed meal | 16.45 | *Aspergillus niger* | 43.07% | Shi et al. [7] |
| Rapeseed meal | 23.79 | *Aspergillus niger* | 30.6% | Tie et al. [76] |
| Rapeseed press cake | 32.1 | *Rhizupus* | 15.9% | Lucke et al. [77] |
| Rapeseed meal | 64.6 | *Lactobacillus delbrueckii* and *Bacillus subtilis* | 94.62% | Zhang et al. [72] |
| Rapeseed meal | 203.7 | *Bacillus subtilis* and *Actinomucor elegans* | 45.26% | Hao et al. [78] |

## 4. Use of Dietary Rapeseed Meal in Pig Nutrition

### 4.1. Growth Performance and Meat Quality

Due to its high content and well-balanced essential amino acids, RSM is a good protein resource of diets for monogastric animals. Recent research has shown that a small amount of RSM could be used in swine diets without detrimental effects on growth performance and meat quality (Table 4). Do et al. demonstrated that a diet containing 8% RSM has no adverse effects on the growth performance of weaning pigs, which is similar to those reported by Shi et al., who found that adding 10% RSM in the diet has no adverse effects on production performance of finishing pig [7,80]. It has been reported that feeding pigs with diets containing RSM doesn't affect the pork meat quality, but it may decrease the weight gain [81,82]. But, Grabez et al. reported addition of RSM to finishing pig diets increased feed conversation ratio, improved meat coloring, increased sweet tasting metabolites and improved the flavor attributes of meat [83]. These differences in these results might be explained by the different RSM breeds and processing treatments, as well as the different breeds and life-stages of experimental pigs.

**Table 4.** The effects of rapeseed meal on growth performance and meat quality of pig.

| Animal | Source | Results | References |
|--------|--------|---------|------------|
| Weanling pig | Rapeseed meal | No adverse effects on the growth performance with up to 8% rapeseed meal | Do et al. [80] |
| Weaned pig | *Brassica napus* and *Brassica juncea* canola meal | No difference in feed intake, BWG and FCR | Landero et al. [84] |
| Growing pig | Rapeseed meal fermented by *Aspergillus niger* | No adverse effects on performance, when replaced with rapeseed meal up to 10% | Shi et al. [7] |
| Growing pig | Canola/double low rapeseed meal/expeller | No difference in growth performance with rapeseed meal up to 5% | Hansen et al. [85] |
| Growing-finishing pig | Rapeseed meal | No adverse effects on performance, when rapeseed meal was provided up to 9% | Choi et al. [81] |
| Finishing pig | Extracted rapeseed meal and legume plant | No adverse effects on pork quality; Reduce fatness; daily BWG↓ | Zmudzinska et al. [82] |
| Growing-finishing pig | Comercial expeller pressed rapeseed | ADG↓; generally no difference in meat quality with rapeseed meal up to 20% | Skugor et al. [86] |
| Growing-finishing pig | Rapeseed meal | FCR↑; glucose level, lightness and yellowness of meat↓; oxidative stress↓; free amino acids, sweet tasting metabolites and flavor attributes↑ | Grabez et al. [83] |
| Growing-finishing pig | Rapeseed meal | FCR↑; total MUFA↑, SFA and PUFA↓ in the steak cuts↓; modified the microbial balance in the digestive tract | Skoufos et al. [87] |

ADG = average daily gain; BWG = body weight gain; FCR = feed conversation ratio; MUFA = monounsaturated fatty acid; SFA = saturated fatty acid; PUFA = polyunsaturated fatty acid.

### 4.2. Reproduction of Sows

Pregnant sows are usually subject to strict feeding restrictions, which unfortunately cause various problems such as constipation and malnutrition [88]. The reproductive performance of sows fed diets supplemented with RSM are included in Table 5. Grela et al. found that the addition of 4–9% share of a fermented RSM component in gestation and lactation sow diet, respectively, improved the production parameters (litter size and litter weight), changed the microbiological composition of the digestive tract contents in pregnant gilts and reduce the severity of diarrhoea and mortality of the offspring [89]. This may be due to the fact that the fermentation process enriches the diet with enzymes, vitamins and short-chain fatty acids, thereby stimulating the gut environments of pig. Meanwhile, RSM contains crude fiber and adding an appropriate amount of fiber to the diet can prevent constipation and increase satiety of the sow. Quiniou et al. reported that sows fed diets with 10% RSM over three reproductive cycles, piglet weight at birth and litter weight gain were not affected [90]. Nevertheless, researchers also found that the addition of 4–9% fermented RSM in pig diet could increase litter size and weight, simulate the immune system (increase LYM counts and IgG titres in the blood plasma), and improve the nutrient digestibility [89,91,92]. These enhancements in pigs may be attributed to fermentation, which could reduce the content of anti-nutrients, improve the gut bacteria structure and promote the nutrients absorption of sows. Taken together, RSM has the potential to improve the reproductive performance of sows, but the processing treatment and applied RSM level in feed need to maintain considerable flexibility.

**Table 5.** The effects of rapeseed meal on reproduction performance of sow.

| Sources | Appending Proportion (%) | Main Results | References |
|---|---|---|---|
| Rapeseed meal | 10% | Not affect piglet weight at birth or weaning, survival and litter weight gain | Quiniou et al. [90] |
| Rapeseed meal | 12% | No detrimental effects on reproductive performance and growth their progeny | Park et al. [93] |
| Rapeseed meal | 6% | No detrimental effects on growth and production | Bowland and Hardin [94] |
| Canola meal | 30% | Gut lactic acid bacteria↑; sow body weight and plasma urea nitrogen↓; No adverse effects on milk composition and nutrient digestibility | Velayudhan et al. [59] |
| Rapeseed press cake Fermented rapeseed meal | 8~14% 4~9% | Body weight of piglet↑; piglet growth rate↑; Stimulate immune and antioxidant system; | Hanczakowska et al. [95] Czech et al. [92] |
| Fermented rapeseed meal | 4~9% | Litter size and litter weight↑; nutrient digestibility↑; maleficent bacteria↓ | Grela et al. [89] |
| Fermented rapeseed meal | 4~9% | Plasma content of Ht, Hb, RBC and mineral↑; plasma content of total cholesterol and triacylglycerols↓; liver enzyme activity↓ | Czech et al. [91] |

Ht = haematocrit; Hb = haemoglobin; RBC = erythrocyte count.

## 5. Conclusions and Future Perspective

Although considerable progress (heat treatment, microbial fermentation, and enzymolysis) has been taken in recent years, some questions still remain unclear.

It is difficult to improve the effectiveness of fermentation such as finding the most suitable temperature and time and optimum ratio of bacteria/enzyme.

Modern methods for processing RSM are not yet perfect.

The metabolic characteristics of probiotics in the fermentation process need more exploration.

To better answer these practical questions, more studies are still needed to be done to fully utilize RSM in swine diets.

**Author Contributions:** H.C., X.L., Q.X., L.T. and N.L.: Literature collection, H.C. and F.Z. Writing-Original draft preparation. X.J., J.C. and J.W.: Writing—Reviewing and Editing. B.T. and X.M.: Funding acquisition. All authors have read and agreed to the published version of the manuscript.

**Funding:** This research was supported by the National Key R&D Program of China (2021YFD1301004), the Hunan Provincial Natural Science Foundation of China (2021JJ30318), the "Open Project Program of Key Laboratory of Feed Biotechnology, the Ministry of Agriculture and Rural Affairs of the People's Republic of China", and the China Agriculture Research System of MOF and MARA, Earmarked Fund for China Agriculture Research System (CARS).

**Institutional Review Board Statement:** Not applicable.

**Informed Consent Statement:** Not applicable.

**Data Availability Statement:** Not applicable.

**Conflicts of Interest:** We declare that there is no conflict of interest.

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
