# Peer review of "Rapeseed Meal and Its Application in Pig Diet: A Review"

_agriculture, doi:10.3390/agriculture12060849_

Round 1

Reviewer 1 Report

Manuscript ID: agriculture-1714907
Title: Rapeseed meal and its application in pig diet: A review

The article provides useful information. The manuscript needs some revisions, because there are some aspects of the work that should be corrected and improved. Please, review the following recommendations:

(1) In each page: "Agriculture 2021, 11, x FOR PEER REVIEW " , update to 2022

(2) Improve the Abstract section

(3) Improve the introduction section, providing the problem

(4) Lines 43, 52, 58, 65: Change "amino acid " to "amino acids"

(5) Revise to the journal's system for writing references, for examples : Lines 62, 126, 210, 217, 261, 266: Not compatible with the journal writing system " Salazar-Villanea et al. (2016) ", " Velayudhan et al. (2017) ", "Maison et al. (2015) " "Rodrigues et al. (2017)", " Do et al. (2017) " and " Grabez et al. (2020) ", Please check

(6) Line 100: Change "Mineral and Vitamin " to "Minerals and vitamins "

(7) Line 125: Add "the" before "palatability "

(8) Line 136: Change "impairing " to "impairs"

(9) Line 141: Change "grain " to "grains"

(10) Line 210: Add "the" before "addition "

(11) Line 261: Add "a" before " diet "

(12) Line 266: Change " additon " to " addition "

(13) Line 277: Change " Research on reproductive performance of sows fed diets with RSM " to " The reproductive performance of sows fed diets supplemented with RSM "

(14) Line 288: Change " in addition " to " the addition "

(15) Line 289: Change " simulated " to " simulate"

(16) Please revise journal style

(17) Update the references, one reference in 2022 is not enough

Reviewer 2 Report

This review paper is focused on rapeseed meal and its application in pig diet. The composition of the feed mixture influences the successful production of pig, which is important from the point of view of economy and efficiency of breeding. The results of the obtained review were properly described. The strength of the study is that in the authors described not only the positives but also the negatives feeding of rapeseed meal. In the case of negatives, they suggested possible solutions to the problems that arose. They also described the impact of feeding rapeseed meal on pig production and reproduction, which is very important from the farming economy.

Reviewer 3 Report

Dear authors,

I read your review with great interest.

However, I do recommend some restructuring the sentences for better clarity. The structure of review is appropriate.

The literature used is up to date.

Line 60 - 62 : please rewrite for better clarity as it is difficult to understand.

Line 217 - 219: do you mean adding phytase in the feed of growing pigs?

Line 233: The term microbial flora is not in the use anymore.. maybe microbial community?

256: Use of dietary rapeseed meal in pig nutrition is better.

 I miss one overall conclusion in the review not just unanswered questions.

Round 2

Reviewer 3 Report

Dear all,

I went through the corrections and recommend accepting the manuscript in the present form.
